# Frequency and Quantity of Egg Intake Is Not Associated with Dyslipidemia: The Hellenic National Nutrition and Health Survey (HNNHS)

**DOI:** 10.3390/nu11051105

**Published:** 2019-05-17

**Authors:** Emmanuella Magriplis, Anastasia-Vasiliki Mitsopoulou, Dimitra Karageorgou, Ioanna Bakogianni, Ioannis Dimakopoulos, Renata Micha, George Michas, Michail Chourdakis, George P. Chrousos, Eleftheria Roma, Demosthenes Panagiotakos, Antonis Zampelas

**Affiliations:** 1Department of Food Science and Human Nutrition, Agricultural University of Athens, Iera odos 75, 11855 Athens, Greece; emagriplis@aua.gr (E.M.); av.mitsopoulou@gmail.com (A.-V.M.); dimitra1807@gmail.com (D.K.); ioanna_bakogianni@hotmail.com (I.B.); ioannis.dimakopoulos@gmail.com (I.D.); renata.micha@tufts.edu (R.M.); gvmichas@gmail.com (G.M.); 2Friedman School of Nutrition Science and Policy, Tufts University, Boston, MA 02215, USA; 3Department of Cardiology, “Elpis” General Hospital of Athens, 11522 Athens, Greece; 4Medical School, Aristotle University of Thessaloniki, University Campus, 54124 Thessaloniki, Greece; mhourd@auth.gr; 5First Department of Pediatrics, Medical School, National and Kapodistrian University of Athens, Mikras Asias 75, 11527 Athens, Greece; chrousos@gmail.com (G.P.C.); roma2el@otenet.gr (E.R.); 6Department of Nutrition and Dietetics, School of Health Science and Education Harokopio University, Athens, Eleftheriou Venizelou 70, 17676 Athens, Greece; dbpanag@hua.gr

**Keywords:** egg consumption, dyslipidemia, egg frequency, egg quantity, hypercholesterolemia

## Abstract

Background: Gaps remain on the safety of egg intake on cardiovascular health, setting the study’s aim to investigate the association between quantity and frequency of egg consumption, with established dyslipidemia. Methods: Study participants (*N* = 3558, 40.3% males) included individuals from the Hellenic National and Nutrition Health Survey (HNNHS), of national representation. Quantity and frequency of egg consumption was determined. Minimally adjusted, multivariable logistic and linear analysis were used to assess egg consumption and dyslipidemia. Results: The more frequent egg consumption compared to no or rare egg consumption significantly decreased the odds of dyslipidemia in the minimally adjusted (Odds Ratio (OR) for frequency: 0.83; 95% Confidence Interval (CI): 0.752, 0.904; OR for quantified frequency: 0.87; 95% CI: 0.796, 0.963) and the fully adjusted models (OR for frequency: 0.80; 95% CI: 0.718, 0.887; OR for quantified frequency: 0.85; 95%CI: 0.759, 0.945). Level of serum cholesterol and LDL-c were significantly lower with higher frequency and quantified frequency of egg consumption in all models. Conclusion: Eggs do not increase the risk of dyslipidemia and can be consumed as part of a healthy diet that is high in fiber and low in saturated fat, without excessive energy intake, by all individuals.

## 1. Introduction

Cardiovascular disease (CVD) remains the leading cause of morbidity and mortality worldwide [1,2], but it is well documented that prevention is effective. By eliminating health risk behaviors, low CVD risk profile can be maintained from young adulthood to middle age [3], and CVD can be prevented by at least 80% [4]. A low CVD risk profile is obtained by maintaining optimal levels of all established modifiable CVD risk factors, primarily proposed by Stamler et al. in 1990 [5], most of which are highly associated with diet. 

Diet is a highly modifiable risk factor encompassed in “lifestyle” factors and makes the largest contribution to CVD risk mortality at the population level across Europe [2]. It is an important determinant of hypercholesterolemia, including total serum cholesterol and Low Density Lipoprotein cholesterol (LDL-c), both of which are well accepted intermediate biomarkers for CVD [6,7] in all age groups. The effect of dietary cholesterol intake to hypercholesterolemia, however, remains controversial and the need for specific nutritional guidelines for cholesterol limits or intake of specific food intake remains debatable, although some large health organizations such as the British Heart foundation [8] and the American Diabetes Association [8] have lifted specific limits, based on recent findings [9,10,11,12], and Dietary Guidelines for Americans do not report specific levels, but continue to advise on low dietary cholesterol intake while consuming a healthy diet [13].

Egg consumption has been a topic of great interest in the past decades, since eggs are nutrient dense food, low in saturated fats but are a good source of cholesterol (egg-yolk), hereby raising the debate about relationships among egg consumption, dyslipidemia and, hence, risk of CVD [9,11,12,14,15]. Some studies have shown modest contribution to plasma (LDLc) levels [11] while others have reported elevated levels of inflammatory indices in animal models upon egg-yolk intake [16] and an increase in vascular inflammation, oxidative stress [2] and 4-h postprandial LDLc increase levels, upon consumption of a high fat-high cholesterol meal [14]. Based on the latter, the authors suggested that egg consumption increases the risk of cardiovascular events, although the diet provided was also high in fat in addition to the cholesterol content. Recent results from a large randomized controlled trial, reported that consumption up to four eggs per week was unrelated to CVD in high risk individuals [12], while results from the Physician’s Health Study showed no effect with infrequent egg consumption, but a positive relationship with mortality, especially in diabetic individuals [17]. A recent meta-analysis showed that Coronary Heart Disease (CHD) does not appear to be associated with consumption of up to one egg per day and may even contribute to a total risk stroke reduction [15], although most studies were small. 

Gaps, therefore, remain on the safety of egg intake on CV health, setting the aim of this study, which was to investigate the association between quantity and frequency of egg consumption, with established hypercholesterolemia, among a Hellenic national representative adult sample. A secondary aim included the examination of short-term egg consumption on serum cholesterol levels, upon adjusting for multiple factors. 

## 2. Methods

### 2.1. Study Design and Subjects

Study participants included individuals from the Hellenic National and Nutrition Health Survey (HNNHS), of national representation. The study was carried out from 1 September 2013 to 31 May 2015, and surveyed non-institutionalized civilians of all ages, living in Greece. Stratification was performed according to: (a) geographical density criteria by Greek region (7 regions), as provided by the Hellenic Statistical Authority; (b) age group; and (c) gender distribution. All non-institutionalized adults and non-pregnant or lactating women (40.8% males), residing within Greece, were included in the study. Details on methodology have been published elsewhere [18]. 

Individuals with missing 24-h recalls as well as extreme over-reporters (defined as individuals reporting >6000 kcal/day) were excluded from the analysis. A remaining 3558 participants were enrolled (40.3% males). The survey included a standardized in-home interview and a physical examination in a mobile examination center. Measurements from a total subsample of 1051 individuals (29.5% of the total adult sample included), were obtained on fasted serum lipids (cholesterol, LDL, and HDL (mmol/L)), blood pressure (mmHg), body weight (kg) and height (m).

All work was carried out upon obtaining individual consent and approval by the Ethics Committee of the Department of Food Science and Human Nutrition of the Agricultural University of Athens and by the by Hellenic Data Protection Authority (HDPA).

### 2.2. Clinical, Dietary and Anthropometric Data

Data were collected by Computer Assisted Personal Interview (CAPI) by experienced study clinicians. Individuals were asked to report their lipid, diabetes and hypertension status and information obtained was evaluated based on medication use and medical history. Individuals were categorized with dyslipidemia based on ICD-10 diagnosis codes or if they had been diagnosed at least once in the past by a clinician, as having high cholesterol or high triglycerides or if they were on lipid lowering medications, as is common in epidemiological studies. Among the subgroup measured, individuals were classified with dyslipidemia if individuals were on lipid lowering medications or had any or a combination of the following: high cholesterol (>5.2 mmol/L), high triglycerides (>1.7 mmol/L), high LDL-cholesterol (>3 mmol/L), and low HDL-cholesterol (<1.03 mmol/L).

### 2.3. Egg Quantification

Information on dietary intake was obtained from each participant, with two non-consecutive 24-h recalls using the technologically validated USDA Automated Multiple-Pass Method (AMPM) [19], 8–20 days apart, and the use of age-specific food atlases. Details on the 24-h recall methods have been recently published [20]. Information on any egg intake, eaten as a whole, partly or in recipes was assessed. The total egg intake in grams per individual over the average of two days was calculated. Total energy, total fat, dietary cholesterol and saturated fat and fiber, were calculated using the Nutrition Data System for Research (NDSR)-USDA based system, and the Greek food composition tables when needed. 

### 2.4. Egg Frequency

The frequency of egg intake was also computed and was used for sub-analysis to account for quantity over time (egg intake in 24-h recalls × frequency of reported egg consumption). Possible responses included “never”, “1 to 3 times per month”, “once a week”, “2–4 times a week”, “5–6 times a week”, “every day”, “2 to 3 times a day”, “4 to 5 times a day”, and “5 to 6 times a day”. 

### 2.5. Egg frequency X Quantity

Frequency of egg consumption was translated into servings per day by dividing mean reported frequency by the total days (for example 1 to 3 times per month: mean 2 times per month divided by 30 = 0.067 servings per day, and once a week = 1/7 = 0.143 servings/day). This was then multiplied by the grams that were quantified from the two 24-h recalls, to obtain a relative intake over time, and decrease variability. 

### 2.6. Clinical and Anthropometrical Data

Cardiovascular risk factors collected, in addition to presence of hypercholesterolemia and/or hypertriglyceridemia, included information on age, smoking status, physical activity, educational level, weight, height, presence of hypercholesterolemia and or hypertriglyceridemia, presence of hypertension and presence of diabetes. Smokers included occasional and current smokers, and physical activity was based on the International Physical Activity Questionnaire (IPAQ) adapted for adults and elderly [21] and categorized as “sedentary”, “light”, “moderate” or “high”, based on the IPAQ score. Educational level was categorized into “low” (<6 years of schooling), “moderate” (up to 12 years of schooling) and “high” (>12 years of schooling; college or higher degree of education). Body weight (kg) and height (m) were used to calculate Body Mass Index (BMI) given by the equation weight/height^2^ (kg/m^2^).

### 2.7. Data Management

In the analysis, multiple variables had to be accounted for to decrease confounding. Adding too many variables in a model can lead to over-adjustment and increases the chance for collinearity and too many pairwise correlations between the variables need to be considered. Therefore, health related variables including, BMI, presence of hypertension, diabetes, age, IPAQ score, and total energy intake were entered in a polychoric correlation matrix for models to decrease the number of variables into specific components. This was preferred to the usual principal component analysis since the data included were a mixture of binary (e.g., hypertension: yes/no), and continuous (e.g., BMI can take any value). Polychoric correlations assume the variables are ordered measurements of an underlying continuum, based on maximum likelihood, and can range from −1 to 1 inclusive and measure the strength and direction of the association between two variables. Once the polychoric correlation matrix was obtained, an exploratory factor analysis using the matrix as input, rather than the raw variables, was performed and specific components were derived based on the degree of variation explained (which is based on the eigen value, the sum of squares of all distances for the best fit line and represents the proportion of variation that each component accounts for). No components were correlated with each other. The goal was to obtain the simplest interpretation, hence select the minimum number of components that explain a large portion of the variation.

Interpretation of the components was based on findings which variables were most strongly correlated with each component, based on their loadings (loadings >0.3 in absolute terms add to the component, >0.5 was deemed overall important). The variables that were correlated were those that the specific component was explained by.

### 2.8. Statistical Analysis

Baseline variable were stratified according to the participants’ lipid status, to examine significant differences. Where significant differences were found, variables were entered in the multivariable models to address potential confounding. Variables addressed included BMI, age, sex, total energy and fiber, educational status, physical activity, hypertension and diabetes. Sex and smoking were also addressed due to a priori knowledge of potential confounding. Sub-analysis by frequency of egg consumption was not performed since only 4.3% of individuals with dyslipidemia reported over 5 times per week intake (low power). 

Continuous baseline data are presented as mean (sd), skewed (fiber) as median (25th–75th percentile range) and the 75th–95th percentile was given for eggs (as per 24-h recall) due to high left skewness. Frequencies (%) were used to show categorical variables. Independent t-test, Mann–Whitney and chi-squared tests were used, respectively. Bar graphs were derived to explore relationship between average reported egg intake and frequency of intake by lipid status. The distribution of mean serum cholesterol (with 95% CI) by frequency of egg consumption, and pairwise mean comparison with Bonferroni correction, was used to examine distributional differences by intake. Minimally adjusted (only for sex) and multivariable logistic and linear regression were used for binary (presence or absence of dyslipidemia), and continuous variables (total cholesterol, HDL-c, etc.), respectively. Skewed variables (e.g., egg quantity) were log transformed. The first model included the health component from the polychoric analysis (KMO: Kaiser–Meyer–Olkin measure of sampling adequacy was derived), the second model included the behavior component (shown in Table 1), and the third multivariable model included all previous and educational level. All *p*-value estimates were based on two-sided tests. STATA 14.0 (StataCorp, Texas ltd.) statistical package was used for the analysis. 

## 3. Results

Baseline variable differences stratified by lipid status (21% subjects with dyslipidemia) were found in age, total energy and fiber intake, BMI, educational and physical activity level, and presence of hypertension and diabetes (Table 2). In the subsample measured, all fasted cholesterol levels (total, HDL-c and LDL-c) and glucose significantly differed. 

Reported mean egg consumption (grams per day) did not differ between those with dyslipidemia compared to those without, but a significant difference was observed in frequency of reported consumption. In general, significant more individuals (irrespectively of the presence of dyslipidemia) reported consuming more than five eggs per week compared to none (*p* for all < 0.001). Only 5% of the participants however reported consuming one egg per day (2.5%) or more than one egg per day (2.5%). Significantly more dyslipidemic subjects reported consuming one egg per week and 2–4 eggs per week compared to those reporting no or rare consumption. 

The relation of the 75th percentile of egg intake reported from the two 24-h recalls, over frequency of intake and lipid status is shown in Figure 1. Overall, those reporting higher frequency of intake had also reported a higher consumption during the 24-h recall (excluding dyslipidemics reporting 2–4 times a week). 

In Figure 2, mean levels of cholesterol with 95% CIs are depicted, showing the distribution of cholesterol levels according to reported frequency of egg consumption. A significantly higher mean serum cholesterol was measured in individuals consuming eggs 1–3 times per month and 2–4 times a week or >5 times a week.

Two principal components were identified from the polychoric analysis, as observed in Table 1. The first component explained 38% of the variation and included BMI, which was positively correlated with being hypertensive, diabetic and older age (Health component). In the second component, BMI (15.2% of variation) was positively correlated with total energy intake and smoking, and negatively with iPAQ score (behavior component). A total KMO of 0.78 resulted, showing sampling adequacy. 

The presence of dyslipidemia was significantly lower for those consuming eggs more frequently compared to no or rare egg consumption in the minimally adjusted only for sex logistic regression and in all other multivariable models (Table 3). When adjustment for all variables was performed, the odds of having dyslipidemia was from 13% to 20% lower (Odds Ratio (OR): 0.80; 95% CI: 0.759–0.945).

Level of serum cholesterol was significantly lower with higher quantity of egg consumption with cholesterol level decreasing by 0.11 mmol/L (95% CI: −0.187, −0.009) with more frequent egg consumption in the minimally adjusted model (according to Taylor expansion for a small change in the predictor variable, we can approximate the difference in the expected mean of cholesterol, by multiplying the coefficient by the change in egg intake or for larger number b-coefficient × log(y)). This remained significant in all models, including the fully adjusted with cholesterol levels being −0.08 lower (95% CI: −0.149, −0.009). Frequency and quantity × frequency of egg was not significantly associated with cholesterol levels in the fully adjusted models (Table 3).

LDL-c was significantly lower in all three situations and in all models, as shown in Table 3. LDL-c levels were −0.07 (−0.138, −0.009), with higher frequency of consumption in the fully adjusted model, −0.15 (−0.239, −0.070) lower for each gram of egg consumed, and −0.08 (−0.147, −0.155) when quantity and frequency were accounted for. No effects were found in any model and with any reported or derived consumption with HDL (data not shown). 

## 4. Discussion

In this study, egg consumption was found to be inversely related to dyslipidemia. The study adds to current knowledge on cholesterol intake through egg consumption and subsequent risk of dyslipidemia, since it examined quantity, frequency and the quantification of egg frequency consumption. The odds of dyslipidemia were lower with increased frequency of egg consumption and frequency in relation to quantity of intake adjusting for age, sex, educational level, health weight and behavior status. Quantity of egg consumed over the two diet recalls was not associated with lipid status, potentially underlining the need to address long-term food intake variability. In a subsample of the population however, where 24-h recalls were in close proximity with the blood collection, an inverse association was found, in most models. All results are based on the fact that, in this study, the majority of the population (95%) consumed <5 eggs per week and the 95th percentile of egg quantity per day was equivalent to one egg (~55 g). 

Eggs are a nutrient dense food source providing essential macro- and micronutrients, while at the same time remaining a moderate calorie source [22]. In comparison to eggs, most food items high in cholesterol are also high in saturated fats. This can partly explain the discrepancies of dietary cholesterol on lipid profile and CVDs, since the negative effects of saturated fat on LDL-c has been shown [23,24,25]. In addition, a randomized trial in overweight men consuming a carbohydrate-restricted diet and three whole eggs or egg substitute found that the egg consumers responded with no change in LDL-c but with a significant increase in HDL-c levels compared with those not eating eggs [26]. This study showed a decrease in LDL-c among egg consumers and no effect on HDL. The method used of acquiring information on egg consumption (frequency versus quantity of intake) must be evaluated and may shed light to the controversial findings. The relation between presence of dyslipidemia and recent egg consumption, obtained by 24-h recalls, although provides more accurate intake in quantity, may lead to erroneous results, due to the high between-day variation. In this study 24-h recalls, frequency and quantified frequency was used to decrease variability and provide a more accurate estimate over time. 

Based on evidence to date, the relationship of egg consumption or cholesterol intake (from eggs) and serum cholesterol with various CV diseases, differ based on adjustments made. More specifically, studies that adjust for saturated fat in the diet find no adverse effect [12,17,27,28,29]. On the contrary, other studies investigating the effects of dietary cholesterol through animal products or in the context of an overall high fat diet report a negative effect [30], as do studies not adjusting for saturated fat or fiber for saturated fat intake in their final analysis [31,32,33]. In this study, saturated fat intake did not differ between participants with or without dyslipidemia, and had no effect on the results, since the analysis was adjusted for total energy. In a recent study, Zhong et al. found a higher of CVD incident with each additional half an egg consumed, but, when the authors adjusted for total dietary cholesterol intake, the association was nulled [33], strengthening the need to adjust for other dietary factors in order to reduce confounded results. 

Overall egg consumption for specific populations, such as those with type 2 diabetes, however, remains in question since some evidence, although limited, suggest that higher egg consumption by diabetic individuals is significantly associated with a higher risk of coronary heart disease [9,34]. In this study, diabetes, which was included in a health status model, along with BMI and hypertension, was inversely related to egg consumption, although, due to the nature of the study, no further analysis could be performed (low power of discrimination). Some studies have also reported potential association of heart failure with high intakes, of more than one egg per day [29], adding gaps to the potential mechanisms of egg metabolism on health, irrespective of serum total cholesterol, LDL-c and presence of dyslipidemia. 

In the context of this study, eggs were studied in relation to dietary fiber consumption, since dietary fiber acts protectively on lipid profile, decreasing the risk of dyslipidemia. In addition, the effect of egg consumption on serum cholesterol was observed from the reported 24-h recall intake, since lipid measurements were performed within the dietary data collection interval. The association, however, between egg consumption and presence of dyslipidemia can be more accurately examined upon considering quantity and frequency of intake to obtain an overall estimation of intake over time.

This study has some limitations due to its cross-sectional nature. More specifically, as a retrospective study, results cannot implore causation since the exposure and outcome were examined at the same time, hence it lacked temporality. In addition, the majority of the individuals were classified based on interview-based questionnaires and medical history. Although only a small sample of the population had blood measurements performed, the final number remained large to allow for associations to be high in power. 

Whether eggs and egg yolks can be consumed extensively cannot be recommended based on evidence to date, but the actual nutritional value of the egg as a whole, yolk included, needs to be acknowledged.

## 5. Conclusions

According to results from this study, eggs do not increase the risk of dyslipidemia, and can be consumed as part of a healthy diet that is high in fiber and low in saturated fat, without excessive energy intake, by all individuals.

## Figures and Tables

**Figure 1 nutrients-11-01105-f001:**
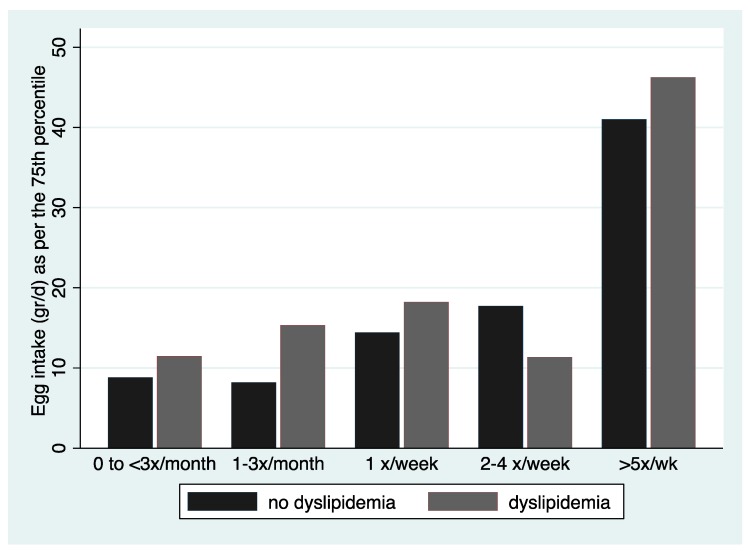
Egg intake (grams/day) in relation to reported frequency of intake by reported lipid status. All individuals were included. Egg intake is reported as per the 75th percentile due to skewed distribution.

**Figure 2 nutrients-11-01105-f002:**
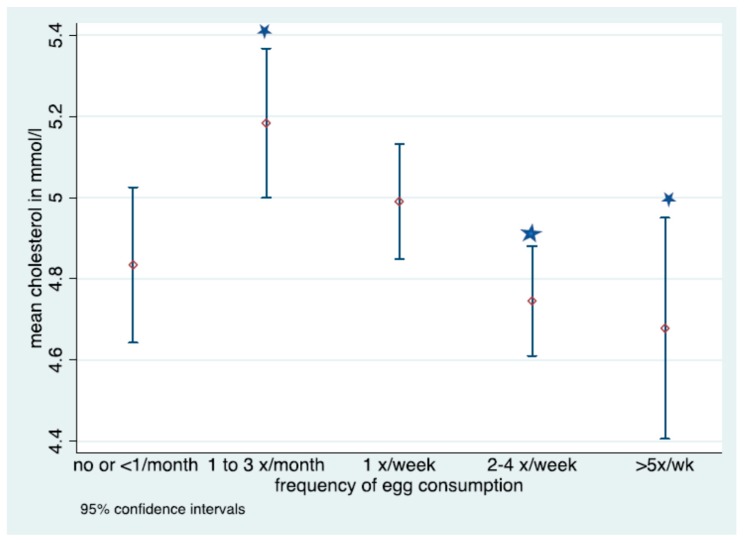
Mean cholesterol and 95% confidence intervals by egg consumption in the measured subpopulation Cholesterol measured in mmol/L. 
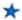
 Denotes significant difference in frequency of consumption; No overlap in CIs denotes statistically significant differences in fasted cholesterol level for those consuming eggs 1–3 times per month and 2–4 times a week or >5 times a week.

**Table 1 nutrients-11-01105-t001:** Principal Components variable loading from Polychoric analysis.

Parameter	Component 1: Health Component	Component 2: Behavior Component
**BMI**	0.369	0.325
**Hypertension**	0.527	
**Type 2 Diabetes**	0.479	
**Smokers**		0.513
**Age**	0.513	
**Total Energy**		0.547
**IPAQ score**		−0.575
**KMO = 0.776**		

KMO: Kaiser–Meyer–Olkin measure of sampling adequacy; BMI: Body Mass Index; IPAQ: International Physical Activity Questionnaire.

**Table 2 nutrients-11-01105-t002:** Baseline and demographic characteristics of the study adult participants by lipid status.

**Parameter**	**Total** **(*N* = 3558)**	**No dyslipidemia** **(*N* = 2807)**	**Dyslipidemia** **(*N* = 751)**	***p*–value** **(by lipid status) ***
Age (years), mean (sd)	43.9 (18.2)	40.2 (17.1)	57.6 (15.5)	<0.001
Sex, % males	1371 (40.3)	1084 (40.4)	287 (40.0)	0.840
Egg intake (grams/day), 75th–95th percentile	13.9–55.2	14.4–56.2	12.6–54.4	0.289
Frequency of egg consumption, *n* (%)				0.001
0–<1 time a month**	259 (10.7)	192 (9.8)	67 (14.5)	-
1–3 times a month	525 (21.6)	411 (20.9)	114 (24.7)	
1 per week**	789 (32.5)	646 (32.8)	143 (31.0)	0.045
2–4 per week **	686 (28.2)	568 (28.9)	118 (25.5)	0.020
>5 per week **	172 (7.1)	152 (7.7)	20 (4.3)	0.002
Total Energy intake (kcal/day), mean (sd)	1971.8 (1141.6)	2016 (1174.4)	1804.7 (992.7)	<0.001
Total SFA (% energy), mean (sd)	12.4 (5.1)	12.5 (5.2)	12.2 (4.7)	0.216
Total fiber, median (25th–75th percentile)	17.1 (10.8–31.8)	17.1 (10.8–33.0)	16.6 (10.6–26.9)	0.0296
BMI (kg/m^2^), mean (sd)	25.5 (4.8)	25.0 (4.7)	27.6 (4.7)	<0.001
Educational Status (level), *n* (%)				<0.001
Low	415 (12.2)	234 (8.7)	181 (25.2)	
Medium	1188 (35.0)	948 (35.4)	240 (33.4)	
High	1793 (52.8)	1496 (55.9)	297 (41.4)	
Smoking status, *n* (%)				0.245
Smokers (daily or occasional)	1147 (33.7)	918 (34.2)	229 (31.9)	
Physical activity status, *n* (%)				<0.001
Sedentary	224 (6.7)	160 (6.1)	64 (9.2)	
Low	463 (13.9)	338 (12.9)	125 (17.9)	
Moderate	1301 (39.1)	1039 (39.5)	262 (37.8)	
Active	1338 (40.2)	1092 (41.5)	246 (35.3)	
Other comorbidities, *n* (%)				
Hypertension ^1^	561 (16.5)	285 (10.6)	276 (38.4)	<0.001
Diabetes	146 (4.3)	66 (2.5)	80 (11.2)	<0.001
**Population subsample**	**Total** **(*N* = 1051)**	**Males** **(*N* = 402)**	**Females** **(*N* = 649)**	***p*–value for sex**
Serum cholesterol (mmol/L), mean (sd)	5.0 (1.1)	4.8 (1.0)	5.6 (1.1)	<0.001
Serum HDL-c (mmol/L), mean (sd)	1.5 (0.4)	1.5 (0.4)	1.4 (0.4)	0.004
Serum LDL-c (mmol/L), mean (sd)	3.0 (0.9)	2.8 (0.9)	3.4 (1.0)	<0.001
Fasted glucose (mmol/L), mean (sd)	5.5 (1.0)	5.5 (0.9)	5.7 (1.1)	<0.001

BMI, Body mass Index; HDL-c, High Density Lipoprotein cholesterol; LDL-c, Low Density Lipoprotein cholesterol; SFA: Saturated Fatty Acids * Significant different at *p* < 0.05 (differences by lipid status); ** Significant differences by frequency of egg consumption between dyslipidemics and non-dyslipidemics (compared to 0–<1 time a month). ^1^ Hypertension defined according to European Society of Cardiology (ESC) >140 systolic blood pressure and/or >90 diastolic blood pressure.

**Table 3 nutrients-11-01105-t003:** Linear and logistic regression analysis of lipid levels and status and egg consumption by reported frequency, quantity, and frequency with quantity.

Models	Frequency of Egg Consumption	Quantity of Egg (g/day) *	Frequency × QUANTITY (g/day) *
OR (95% CI)			
Presence of dyslipidemia **			
Sex adjusted	0.83 (0.752, 0.904)	0.91 (0.809, 1.017)	0.87 (0.796, 0.963)
Multivariable ^1^	0.83 (0.755, 0.915)	0.93 (0.831, 1.055)	0.88 (0.801, 0.974)
Multivariable ^2^	0.83 (0.752, 0.911)	0.92 (0.821, 1.041)	0.88 (0.796, 0.968)
Multivariable ^3^	0.80 (0.718, 0.887)	0.91 (0.798, 1.040)	0.85 (0.759, 0.945)
Linear Regression			
Models	*b*-coefficient (95% CI)	*b*-coefficient (95% CI)	*b*-coefficient (95% CI)
Cholesterol levels ***			
Age & sex adjusted	−0.11 (−0.183, −0.038)	−0.10 (−0.187, −0.009)	−0.11 (−0.190, −0.044)
Multivariable ^1^	−0.10 (−0.178, −0.031)	−0.10 (−0.190, −0.007)	−0.10 (−0.178, −0.028)
Multivariable ^2^	−0.11 (−0.178, −0.031)	−0.10 (−0.193, −0.010)	−0.11 (−0.180, −0.031)
Multivariable ^3^	−0.08 (−0.149, −0.009)	−0.09 (−0.175, 0.004)	−0.07 (−0.143, 0.002)
LDL-c levels ***			
Age & sex adjusted	−0.10 (−0.163, −0.029)	−0.13 (−0.217, −0.052)	−0.12 (−0.187, −0.056)
Multivariable ^1^	−0.09 (−0.160, −0.023)	−0.15 (−0.236, −0.067)	−0.11 (−0.182, −0.047)
Multivariable ^2^	−0.09 (−0.160, −0.024)	−0.15 (−0.239, −0.070)	−0.12 (−0.184, −0.049)
Multivariable ^3^	−0.07 (−0.138, −0.009)	−0.15 (−0.239, −0.070)	−0.08 (−0.147, −0.155)

* Reported frequency of intake (per day) multiplied with average egg intake reported from the 24-h recalls; log transformed to obtain normal distribution. ^**^ Based on information obtained by all individuals; ^***^ based on subsample measured. ^1^ Adjusted for health component (BMI, age, diabetes, and hypertension, as per Component 1). ^2^ Adjusted for behavior (BMI, smoking, total energy intake and activity level, as per Component 2). ^3^ Adjusted for all and educational level. LDL-c: Low Density Lipoprotein-cholesterol.

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
