# Peer review of "Frequency and Quantity of Egg Intake Is Not Associated with Dyslipidemia: The Hellenic National Nutrition and Health Survey (HNNHS)"

_nutrients, 2019, doi:10.3390/nu11051105_

Round 1

Reviewer 1 Report

This study by Fappa et al. investigates the association between dyslipidemia and frequency and quantity of egg consumption in a Greek population. The authors conclude that egg consumption does not increase risk of dyslipidemia, and suggest that egg intake may have a protective effect when combined with a healthy eating pattern. The authors could consider the following points:

Major Limitations:

1.     Dyslipidemia should be defined in the methods section so that it clear what cutoff values were used for total cholesterol and LDL cholesterol. Was hypertriglyceridemia included in the definition or dyslipidemia?

2.     In the methods section (lines 93-95), it says that fasting cholesterol, LDL and HDL were measured in a subsample of the study population. Lines 25-26 state that clinical data collected included the presence of hypercholesterolemia and/or hypertriglyceridemia, presumably on all study participants. This needs clarification. How were hypercholesterolemia identified if cholesterol was only measured in a subsample? Was this data taken from medical records? In Table 2, the data is grouped by dyslipidemia/no dyslipidemia for the entire sample – how was lipemia identified in the subjects that were not tested for fasted lipids?

3.     Table 2 states that the p-value reported is for sex, whereas line 180 states “baseline differences stratified by lipid status were found in…”. Do the p-values reported in Table 2 denote differences between individuals with/without dyslipidemia, or are the differences significant between males and females?

4.     Line 189 states “reported mean egg consumption did not differ”. Please include which groups are being compared within the text.

5.     Two 24-hour recalls may not reflect long-term patterns of food intake. However, the authors have identified this limitation in the discussion section.

6.     Overall, the conclusion that egg consumption is not associated with dyslipidemia is sound. However, in Figure 2, individuals consuming of 1-3 eggs/month have higher cholesterol, whereas circulating cholesterol is the same in those consuming 0 or >1 egg/month compared to higher frequencies of egg consumption. In light of these findings, the statement that egg consumption may have a protective effect may not be justified.

7.     Please describe to the novelty of this study. Numerous studies have examined the relation between egg consumption and circulating cholesterol. How do your results add to the existing knowledge in this field?

Minor Limitations:

1.     It may be of benefit to have the manuscript reviewed by a native English speaker to correct minor errors.

2.     For the frequency of egg consumption in Table 2, does the asterisk indicate a significant difference between the “no dyslipidemia” and “dyslipidemia” groups within a given egg consumption frequency? Please clarify what groups are being compared in the footnote.

Author Response

Comments and Suggestions for Authors

This study by Fappa et al. investigates the association between dyslipidemia and frequency and quantity of egg consumption in a Greek population. The authors conclude that egg consumption does not increase risk of dyslipidemia, and suggest that egg intake may have a protective effect when combined with a healthy eating pattern. The authors could consider the following points:

    Authors: We thank the reviewer for the comments. It has to be noted that the first author is         Magriplis E.

Major Limitations:

1.     Dyslipidemia should be defined in the methods section so that it clear what cutoff values were used for total cholesterol and LDL cholesterol. Was hypertriglyceridemia included in the definition or dyslipidemia?

Authors: Thank you for your comment. We have added the missing information in the paper.  

2.     In the methods section (lines 93-95), it says that fasting cholesterol, LDL and HDL were measured in a subsample of the study population. Lines 25-26 state that clinical data collected included the presence of hypercholesterolemia and/or hypertriglyceridemia, presumably on all study participants. This needs clarification. How were hypercholesterolemia identified if cholesterol was only measured in a subsample? Was this data taken from medical records? In Table 2, the data is grouped by dyslipidemia/no dyslipidemia for the entire sample – how was lipemia identified in the subjects that were not tested for fasted lipids?

Authors: Details on dyslipidemia classification are now written in Methods section, as the reviewer correctly questions. Data on all individuals were collected from computer assisted personal interviews. Information obtained was crosschecked with medications consumed and previous diagnosis by a clinician (medical records). This was performed in all individuals.

For the subsample measured, dyslipidemia was defined based on specific lipid values (as now clearly stated in methods). Analysis of data for dyslipidemia association with egg consumption was performed separately for those measured and those that had “reported”. The latter is specified in Data management section, and in all tables/graphs.

3.     Table 2 states that the p-value reported is for sex, whereas line 180 states “baseline differences stratified by lipid status were found in…”. Do the p-values reported in Table 2 denote differences between individuals with/without dyslipidemia, or are the differences significant between males and females?

Authors: We apologize for the misprint and thank you for the comment. As stated in results and shown in the actual table, baseline differences were stratified by lipid status. The title has now been corrected.

4.     Line 189 states “reported mean egg consumption did not differ”. Please include which groups are being compared within the text.

Authors: The groups compared (those with dyslipidemia compared to those without) have now been included.

5.     Two 24-hour recalls may not reflect long-term patterns of food intake. However, the authors have identified this limitation in the discussion section.

    Authors: Thank you. The authors would like to underline that although two 24-hr recalls do not reflect long term intake, it can reflect measured serum lipids since these were performed within the dietary data collection.   Also, long term frequency of intake (response to comment #7 as well), was used alone and in correlation with mean intake from the 24-hour recalls, in order to estimate longer term quantification and ameliorate data of long-term intake (account for between day variation in consumption and decrease estimations from reported frequencies). This is one of the novelties of this study. Also, this paper has used a specific statistical analysis (based on a polychoric correlation matrix) to account for the multifactorial nature of dyslipidemia, hence addressing all major confounders, without collinearity, as many studies have omitted to perform (including dietary fiber, energy…)

6.     Overall, the conclusion that egg consumption is not associated with dyslipidemia is sound. However, in Figure 2, individuals consuming of 1-3 eggs/month have higher cholesterol, whereas circulating cholesterol is the same in those consuming 0 or >1 egg/month compared to higher frequencies of egg consumption. In light of these findings, the statement that egg consumption may have a protective effect may not be justified.

Authors: The authors have changed the statement as the reviewer suggests, to address results with greater accuracy. The sentence now reads: According to results from this study, eggs do not increase the risk of dyslipidemia and can be consumed as part of a healthy diet, that is high in fiber and low in saturated fat, without excessive energy intake, by all individuals.” The authors, however, would like to address the specific figure and state that although those with 0-1 egg/month had lower mean values compared to 1-3 times/month, in both cases consuming >2 eggs/week resulted to lower mean cholesterol levels.

7.     Please describe to the novelty of this study. Numerous studies have examined the relation between egg consumption and circulating cholesterol. How do your results add to the existing knowledge in this field?

Authors: The reviewer is correct. Numerous studies have been performed, with each either using 24-hr recall data, frequency of consumption and short-term food records. Other to the fact that controversies remain which may be affected by the type of consumption data collected (as addressed in the paper), this study addressed this issue by examining quantity, frequency as well as the quantification of frequency (see the response to question 5 also). We added the following sentence at the first paragraph of the discussion: “The study adds to current knowledge on cholesterol intake through egg consumption and subsequent risk of dyslipidemia, since it examines quantity, frequency as well as the quantification of egg frequency consumption”.

Minor Limitations:

1.     It may be of benefit to have the manuscript reviewed by a native English speaker to correct minor errors.

    Authors: We addressed your comment and gave the paper to a second native speaker (the first author is a native speaker, Canadian origin), and some corrections were made.

2.     For the frequency of egg consumption in Table 2, does the asterisk indicate a significant difference between the “no dyslipidemia” and “dyslipidemia” groups within a given egg consumption frequency? Please clarify what groups are being compared in the footnote.

    Authors: In table 2, there is no asterisk, therefore, the authors presume that the reviewer referred to figure 2. In this case, the title was modified to correctly reflect the figure, which was mean fasted cholesterol values by frequency of consumption, in the subpopulation measure.  

Reviewer 2 Report

To:

Editorial Board

Nutrients

Title: “Frequency and quantity of egg intake is not associated with dyslipidemia: the Hellenic National Nutrition and Health Survey (HNNHS)”

Dear Editor,

I read this manuscript dealing with the role of egg intake in counteracting dyslipidemia.

I think that the paper is good and well written. The authors considered the evaluation of 3558 individuals (751 dyslipidemic). Although this is a retrospective study, based on a standardized survey and phone call, the statistics overcome these limitations. The multivariate regression analysis succeeded in better evaluating the relationship between egg intake and dyslipidemia. Therefore, the scientific background of the text is supported by statistical analysis. At least, the authors can briefly outline the retrospective nature of the analysis and the use of a questionnaire into a dedicated limitation section.

Author Response

I think that the paper is good and well written. The authors considered the evaluation of 3558 individuals (751 dyslipidemic). Although this is a retrospective study, based on a standardized survey and phone call, the statistics overcome these limitations. The multivariate regression analysis succeeded in better evaluating the relationship between egg intake and dyslipidemia. Therefore, the scientific background of the text is supported by statistical analysis. At least, the authors can briefly outline the retrospective nature of the analysis and the use of a questionnaire into a dedicated limitation section.

Authors: We thank the reviewer very much for the kind words and positive feedback. As per the reviewers' recommendation, we have added a brief more specific statement on the lack of temporality due to the fact that cross-sectional studies examine exposure and outcome at the same time. Also, a comment that the majority of the population were classified based on interview-based questionnaires, has also been included.  

Round 2

Reviewer 1 Report

Thank you for your responses to the previous questions. Overall, responses to the questions are adequate and many changes indicated in the previous review have been made. However, there are some issues that were pointed out in the previous review that have not been fully addressed:

1.     The title of Table 2 has been corrected to indicate that baseline differences were stratified by lipid status. However, the table heading still states “p-value for sex”. Should this read “p-value for lipid status”?

2.     The authors have modified the conclusion to remove the statement that egg consumption may have a protective effect. The conclusion statement accurately represents the results observed in this study; however, they may consider modifying this statement in the abstract as well.

3.     In Table 2, under “frequency of egg consumption” this reviewer was referring to the asterisks following 0- <1 time a month, 1 per week, 2-4 per week and >5 per week. Presumably these asterisks indicate significant differences between the “no dyslipidemia” and “dyslipidemia” groups within a given frequency of egg consumption. For clarity, the authors may consider explaining this in the footnote, or providing a separate p-value for each frequency of egg consumption.  The footnote only state what the asterisk means statistically - not does not  clearly define what the comparisons were in terms of data groups - it is assumed it is between the no dyslipidemia and dyslipidemia group, but this needs to be clear and explicit.

Author Response

Thank you for your responses to the previous questions. Overall, responses to the questions are adequate and many changes indicated in the previous review have been made. However, there are some issues that were pointed out in the previous review that have not been fully addressed:

1.     The title of Table 2 has been corrected to indicate that baseline differences were stratified by lipid status. However, the table heading still states “p-value for sex”. Should this read “p-value for lipid status”?

Authors: It is now clearly stated in table 2, that the P-value is for significant differences by lipid status in Table’s column and has also been added in the footnote.

2.     The authors have modified the conclusion to remove the statement that egg consumption may have a protective effect. The conclusion statement accurately represents the results observed in this study; however, they may consider modifying this statement in the abstract as well.

Authors: We sincerely apologize for omitting to change the abstract. It has now been modified.

3.     In Table 2, under “frequency of egg consumption” this reviewer was referring to the asterisks following 0- <1 time a month, 1 per week, 2-4 per week and >5 per week. Presumably these asterisks indicate significant differences between the “no dyslipidemia” and “dyslipidemia” groups within a given frequency of egg consumption. For clarity, the authors may consider explaining this in the footnote, or providing a separate p-value for each frequency of egg consumption.  The footnote only state what the asterisk means statistically - not does not clearly define what the comparisons were in terms of data groups - it is assumed it is between the no dyslipidemia and dyslipidemia group, but this needs to be clear and explicit.

Authors: Thank you for the comment. Specific p-values have now been included and details have been entered in footnote, accordingly.